# Tissue Microenvironment as an Additional Prior for Visual Representation Learning in Histopathology

## Abstract

Self-supervised learning has transformed histopathology by enabling foundation models to learn from vast unlabeled image archives, with methods developed using natural images, such as DINOv2, establishing powerful baselines. We propose augmenting these approaches by incorporating tissue microenvironment structure as an additional prior through semantic masking. We train adversarial mask generators adapted from ADIOS with perceptual reconstruction losses to identify tissue structures, then integrate these semantic masks as augmentations within DINOv2 self-supervised learning pipelines. Using a set of 55 million TCGA histopathology tiles of 224×224 pixels at a resolution of 0.5 $\mu$m/pixel, we pre-train vision transformers to evaluate three augmentation strategies: standard DINOv2 augmentations, mixed (combining standard and semantic masking), and semantic masking only. The mixed augmentation strategy, when used in DINOv2, demonstrates consistent improvements over baseline across four patch-level classification benchmarks (PCam, MiDOG, MHIST, BRACS) and on two slide-level mutation prediction tasks (EGFR in LUAD, FGFR3 in BLCA). Qualitative PCA visualization of patch tokens shows that semantic masks combined with standard augmentations enable a better decomposition of tissue into biologically interpretable components without supervision, with DINOv2-mixed achieving clear separation of cellular structures, vasculature, and stromal elements. Therefore, incorporating domain-specific tissue priors through semantic masking enhances representation learning in self-supervised frameworks, alongside standard augmentations.

## 1 INTRODUCTION

Self-supervised learning has transformed visual representation learning in computational pathology by leveraging vast archives of unlabeled histopathology images. Current approaches predominantly rely on enforcing representational invariances through contrastive learning Chen et al. (2020); He et al. (2020), masked image modeling He et al. (2022); Zhou et al. (2022), and self-distillation frameworks Caron et al. (2021); Grill et al. (2020); Caron et al. (2020). The synthesis of these techniques in DINOv2 Oquab et al. (2023), i.e., combining self-distillation, masked tokens, momentum optimization, cluster assignments, and joint-embedding architectures, has established the de facto backbone for pathology foundation models including UNI Chen et al. (2024), Virchow Vorontsov et al. (2024), REMEDIS Azizi et al. (2023), CTransPath Wang et al. (2022), HIPT Chen et al. (2022), Prov-GigaPath Xu et al. (2024), CHIEF Wang et al. (2024a), and Phikon Filiot et al. (2023b).

However, these methods inherit augmentation strategies from natural image domains that contrast against histopathology's unique characteristics. Histopathology images are small tiles extracted from gigapixel whole-slide images, lacking the foreground-background relationships and perspective cues present in natural images. Instead, they contain hierarchically organized tissue structures where semantic meaning emerges from the spatial arrangement of cellular components such as nuclei, stroma, vasculature, and glands. While previous work has adapted augmentations for stain variations and geometric transformations Tellez et al. (2019); Faryna et al. (2021); Otálora et al. (2022), these approaches treat tissue as homogeneous texture rather than leveraging its inherent semantic structure.

We propose that the tissue microenvironment itself provides a natural augmentation strategy through semantic masking. Unlike random masking, semantic masks can selectively occlude biologically meaningful components, thus forcing models to learn relationships between tissue structures rather than low-level texture patterns. This approach draws inspiration from ADIOS Shi et al. (2022), which learns adversarial masks that create challenging yet learnable occlusions. By training mask generators to identify tissue components and using these masks as augmentations within the DINO framework, we hypothesize that models will develop representations that better capture the hierarchical organization needed for diagnostic tasks.

The main contributions of this paper are:

- We introduce a variant of ADIOS Shi et al. (2022) optimized for separating meaningful sematic structures for the tissue microenvironment (i.e. cell nuclei, cell borders, vascular structures) from histopathology images.

- We show that adding semantic masking augmentations based on the tissue microenvironment to standard DINOv2 training consistently improves representation learning, yielding gains across four patch-level classification benchmarks and two slide-level mutation prediction tasks.

- We show through PCA visualization that adding semantic masks augmentations to DINOv2 training results in better partitioning of tissue into biologically interpretable components without supervision.

## 2 RELATED WORK

**Self-Supervised Learning in Histopathology**: Self-supervised learning (see Balestriero et al. (2023) for a review) has emerged as the dominant paradigm for histopathology foundation models, addressing annotation scarcity in gigapixel whole slide images (WSIs) Filiot et al. (2023a). Current approaches leverage contrastive learning Chen et al. (2020); He et al. (2020), masked image modeling He et al. (2022); Zhou et al. (2022), and self-distillation frameworks Caron et al. (2021; 2020), with DINOv2 Oquab et al. (2023) synthesizing these techniques to establish the de facto backbone for pathology models including UNI Chen et al. (2024), Virchow Vorontsov et al. (2024), and Prov-GigaPath Xu et al. (2024). While domain-specific SSL consistently outperforms ImageNet initialization Kang et al. (2023), these methods critically inherit augmentation strategies from natural images—random crops, color jittering, and geometric transformations—that fail to exploit histopathology's unique hierarchical tissue organization. Previous adaptations address stain variations Tellez et al. (2019); Faryna et al. (2021); Otálora et al. (2022) but continue treating tis-

sue as homogeneous texture rather than leveraging its inherent semantic structure where meaning emerges from spatial arrangements of nuclei, stroma, vasculature, and glands. This fundamental mismatch between augmentation strategy and domain characteristics motivates incorporating tissue microenvironment structure directly into the learning process.

**Unsupervised Segmentation**: Several unsupervised learning approaches currently exist in literature. ADIOS Shi et al. (2022) pioneered the approach of learning adversarial masks that create semantically meaningful occlusions, during representation learning, forcing models to perform complex reasoning rather than exploiting local correlations, demonstrating improvements over some self-supervised learning frameworks. Methods like STEGO Hamilton et al. (2022) operate on pre-trained DINO features, distilling semantic relationships through contrastive losses without modifying the underlying representations. Other approaches include COMUS Zadaianchuk et al. (2023), which uses saliency-guided clustering, CutLER Wang et al. (2023), which applies normalized cuts for instance segmentation, UnSAM Wang et al. (2024b), which uses a divide-and-conquer approach to generate pseudo masks, and Object-centric approaches like Slot Attention Locatello et al. (2020), which structure representations as exchangeable slots. While any of these approaches can produce semantic masks required for the subsequent pretraining, the ADIOS framework was ultimately chosen due to its implementation simplicity, and the flexibility of cheaply generating a desired number of semantic masks per image using a relatively small masking model. The framework also offered the opportunity to study the performance of simultaneously trained representations in future work.

# 3 APPROACH

## 3.1 SEMANTIC MASK GENERATION VIA ADVERSARIAL LEARNING

### 3.1.1 TECHNIQUE

Given an unlabeled dataset of histopathology images $\mathcal{D} = \{x_i\}_{i=1}^N$ where $x_i \in \mathbb{R}^{H \times W \times 3}$, we adapt the ADIOS framework Shi et al. (2022) to generate semantic masks for tissue structures. While ADIOS learns adversarial masks that create challenging occlusions for self-supervised learning, the original framework alone lacks grounding constraints, potentially generating masks that are difficult for the encoder but semantically incoherent. We address this limitation by introducing a reconstruction phase that forces masks to identify coherent tissue structures that can be meaningfully recovered from partial observations.

The framework consists of three components: an encoder $f_\theta$ that learns representations from masked and unmasked images, a mask generator $g_\psi$ that produces $K = 3$ semantic masks (with biological motivations from the primary tissue components in H&E sections: nuclei, stromal regions, and interstitial structures), and a reconstructor $r_\omega$ that grounds the masks by enforcing reconstructibility. Given an input image $x$, the mask generator produces soft masks $M = g_\psi(x) = \{m_1, m_2, m_3\}$ where $m_k = \sigma(g_\psi^k(x)) \in [0, 1]^{H \times W}$.

Per batch iteration, the training alternates between three optimization phases:

**Phase 1 - Contrastive Learning**: The encoder learns invariances between original images and their masked variants. For each image $x$ and its masks $M$, we create masked views $\tilde{x}_k = x \odot (1 - m_k)$ and extract $C$ crops per mask. The contrastive loss maximizes agreement between the original and all masked views simultaneously:

$$\mathcal{L}_{\text{student}} = -\frac{1}{N} \sum_{i=1}^N \frac{1}{|\mathcal{P}_i|} \sum_{p \in \mathcal{P}_i} \log \frac{\exp(z_i^\top z_p / \tau(t))}{\sum_{k \neq i} \exp(z_i^\top z_k / \tau(t))}, \tag{1}$$

where $\mathcal{P}_i$ contains positive indices for anchor $i$, and $\tau(t)$ follows a cosine schedule from 0.2 to 0.05.

**Phase 2 - Reconstruction Grounding**: To ensure masks correspond to meaningful tissue structures rather than arbitrary patterns, we introduce a reconstruction objective. The reconstructor must recover the original image from a hybrid input that combines one mask's content with guidance from others:

$$x_{\text{hybrid}} = x \odot m_1 + (1 - m_1) \odot [m_2 \oplus m_3], \tag{2}$$

where $\oplus$ denotes channel concatenation. The reconstruction loss combines pixel-level accuracy with perceptual similarity:

$$\mathcal{L}_{\text{recon}} = \|r_\omega(x_{\text{hybrid}}) - x\|_1 + \mathcal{L}_{\text{perceptual}}(r_\omega(x_{\text{hybrid}}), x). \tag{3}$$

The perceptual term compares class tokens from intermediate transformer layers of a pre-trained vision transformer base (ViT-B) (weighted 0.1, 0.2, 0.3, 0.4 for layers L/4, L/2, 3L/4, L, L being the total transformer blocks), ensuring semantic coherence beyond pixel matching.

**Phase 3 - Adversarial Mask Learning**: The mask generator creates challenging yet reconstructible masks by optimizing:

$$\mathcal{L}_{\text{mask}} = -\mathcal{L}_{\text{student}} + \alpha \mathcal{L}_{\text{sparsity}}(M) + \mathcal{L}_{\text{cycle}}, \tag{4}$$

where $\mathcal{L}_{\text{sparsity}}(M) = \frac{1}{K} \sum_{k=1}^{K} \frac{1}{\sin(\pi \cdot \bar{m}_k) + \epsilon}$ prevents trivial all-zero or all-one solutions ($\alpha = 0.1$), and the cycle consistency term $\mathcal{L}_{\text{cycle}} = \|r_\omega(x_{\text{hybrid}}) - x\|_1$ ensures masks remain semantically grounded. This combination of adversarial learning with reconstruction constraints produces masks that identify biologically meaningful tissue components rather than arbitrary difficult patterns.

**Baseline ADIOS:** In contrast to the described approach, the baseline ADIOS framework ignores phase 2 in the above approach, and $\mathcal{L}_{cycle}$ in phase 3. Also, in the original ADIOS approach, multiple masks are not generated simultaneously in phase 1, but sequentially. However, we incorporate our phase 1 into the baseline ADIOS framework for any qualitative comparisons.

### 3.1.2 IMPLEMENTATION

**Mask Generator and Reconstructor:** The mask generator $g_\psi$ and reconstructor $r_\omega$ utilize Vision Transformer-tiny (ViT-tiny) encoders connected to U-Net decoders following a single-branch CellViT decoderHörst et al. (2024), incorporating spectral normalization Miyato et al. (2018) for stability during training. For the mask model, the masks are concatenated and passed through a softmax layer so that mask activations are unique, while this is avoided in the reconstruction model. All batch normalization layers are replaced by instance normalization, which contributed the most towards maintaining stable mask generation.

**Encoder:** The encoder $f_\theta$ employs the ViT-B Dosovitskiy et al. (2020) variant with patch size 16. Full architectural details are provided in Section 3.2.2 The contrastive loss projector head is a 2 layer MLP with bottleneck dimension of 256, embedding dimension of 2048.

**Training:** Models are trained using AdamW Loshchilov & Hutter (2017) with base learning rate $5 \times 10^{-5}$ and linear batch size scaling. The mask generator and reconstructor use learning rates of $0.05\times$ and $0.1\times$ respectively, with separate optimizers for each component. Training dataset is the same as described in Section 3.2.2, with total batch size of 1,024. Training employs distributed data parallelism across 4x80GB H100/A100 GPUs with bfloat16 mixed precision Burgess et al. (2019). The mask generator is updated for 40,000 iterations then frozen for use in subsequent self-supervised learning experiments, as extended training leads to mask degradation (see Appendix E).

**Mask Output:** Figure 1 shows semantic masks generated at iteration 40,000. The three masks (visualized as RGB channels) capture distinct tissue components in liposarcoma tissue: muscular architecture (blue channel), nuclear regions (red channel), and stromal elements (green channel). Our reconstruction-grounded approach achieves clearer structural separation compared to baseline ADIOS, particularly for nuclear segmentation.

## 3.2 INTEGRATING SEMANTIC MASKS INTO SELF-SUPERVISED LEARNING

### 3.2.1 SEMANTIC MASKING AS A DATA AUGMENTATION POLICY

In the modified DINO framework, given an input image $x$, we generate $K$ semantic masks $M = g_\psi(x) = \{m_1, ..., m_K\}$ using the frozen mask generator. The teacher network processes two global views of the original image with minimal augmentations: $x_1^t, x_2^t$. The student network processes these same teacher views, plus $K$ globally masked versions $\tilde{x}_k = x \odot (1 - m_k)$ for $k \in \{1, ..., K\}$, and multiple local crops extracted from each masked image. We do not include any semantically guided masking in the patch-level loss based on iBOT Zhou et al. (2022), therefore the patch-masking remains random, and set at a constant of 40% of all patches. We trial three blends of augmentations in this work, which we indicate in table 1.

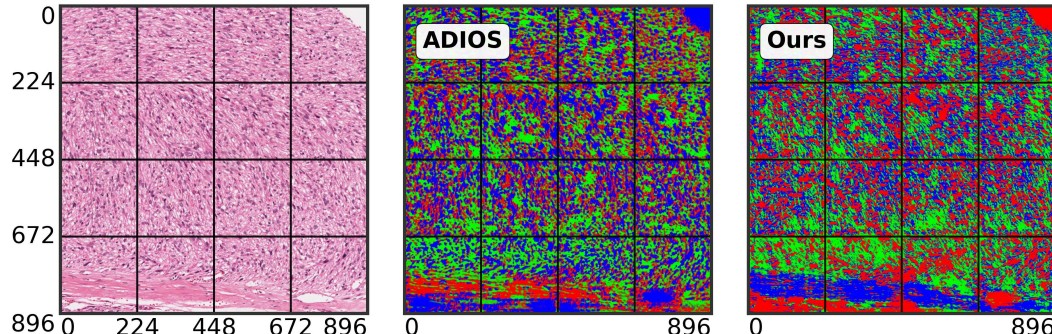

Figure 1: **Semantic mask generation at iteration 40,000.** Liposarcoma tissue (TCGA-SARC) segmented using baseline ADIOS (left) versus our reconstruction-grounded approach (right). Masks are generated using 224×224 non-overlapping patches from the 896×896 image. Three semantic masks visualized as RGB channels.

Table 1: **Training configurations for augmentation strategies.** Settings for global and local crops across Standard, Mixed, and Masked variants for DINOv2. Standard uses conventional augmentations only; Mixed combines standard and semantic masking; Masked uses semantic masking exclusively. Further details provided in Appendix A, and augmentations visualized in Figure 2. See Appendix B.5 for ablation studies including DINOv1 configurations.

| Variant | Global Crops | | Local Crops | |
|---|---|---|---|---|
| | Standard | Masked | Standard | Masked |
| Standard | 2 | 0 | 8 | 0 |
| Masked | 0 | 3 | 0 | 6 |
| Mixed | 2 | 3 | 3 | 3 |

### 3.2.2 IMPLEMENTATION DETAILS

**Main Backbone**: All encoders in this work employ the ViT-B Dosovitskiy et al. (2020) variant with patch size 16, incorporating optimized XFormers attention blocks Lefaudeux et al. (2022), SwiGLU activations Shazeer (2020), LayerScale Touvron et al. (2021), and four register tokens Darcet et al. (2024). The architecture uses post-normalization, learnable position embeddings, stochastic depth Huang et al. (2016), and gradient checkpointing Chen et al. (2016) between the transformer blocks.

**Projector Heads**: The DINO, iBOT, and contrastive loss projector heads are 2 layer MLP with bottleneck dimension of 256, embedding dimension of 2048. A final linear layer of 65536 dimensions is chosen for the DINO and iBOT projector heads, which is weight normalized, and its training is frozen for the first 1250 iterations.

**Training**: For DINOv2 training runs using the frozen mask generator from iteration 40,000, the main encoder is trained for a total of 300K iterations using AdamW Loshchilov & Hutter (2017) optimizer with base learning rate $5 \times 10^{-5}$ incorporating the linear scaling rule of global batch size multiplied by the base learning rate and divided by 256, with warm up from $10^{-6}$ over 12500 iterations, and followed by cosine decay. The gradients are clipped at a gradient norm of 1.0. A weight decay on a cosine schedule from 0.04 to 0.4 is done throughout the training run, and the teacher weight update is done using exponential moving average over the student's weights, with a blending ratio that starts at 0.996 to 1 using a cosine schedule over training iterations. We employ distributed data parallel (DDP) training across GPUs with gradient checkpointing to manage memory consumption, with special care taken to avoid interference between the DDP process and the gradient checkpointing within the PyTorch framework (Paszke, 2019). We used 4×80GB A100/H100 GPUs for training, using the bfloat16 format Burgess et al. (2019) for automated mixed-precision training.

**Encoder Training Specifics**: Hyperparameters follow from that used in Chen et al. (2024). The teacher temperature warm-up occurs over 37500 iterations. The momentum of the teacher's center

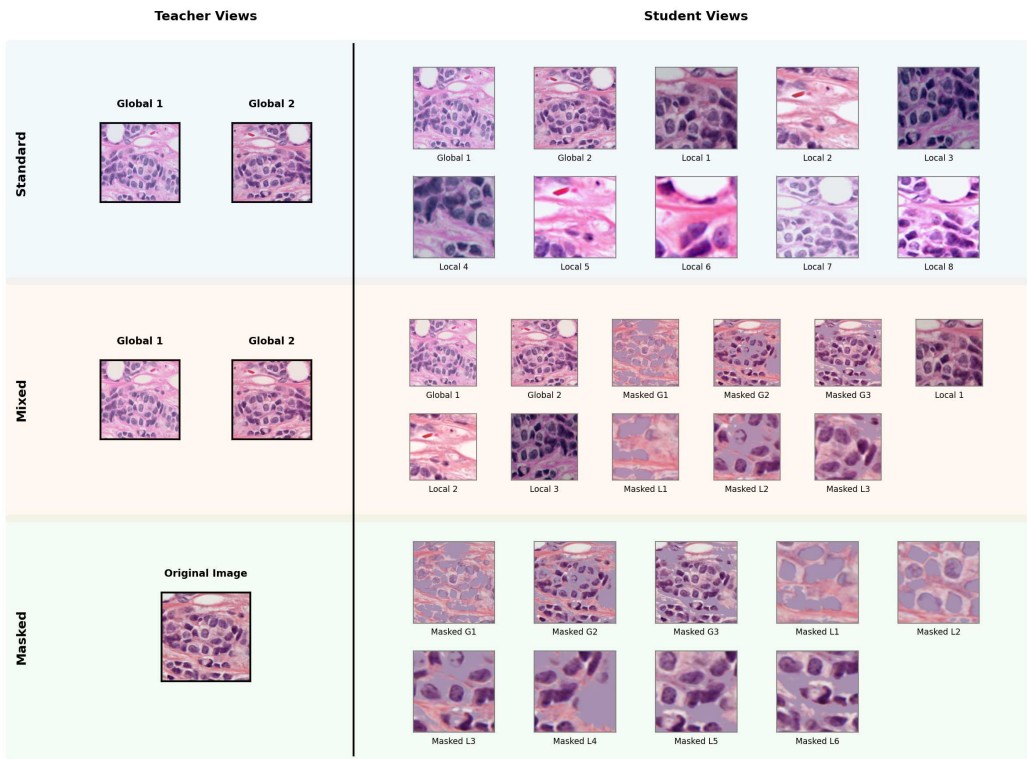

Figure 2: **Visual comparison of augmentation strategies for DINOv2 training.** Left column shows teacher network inputs, right column shows student network inputs. **Standard**: Teacher processes 2 global crops; student processes same 2 global crops plus 8 local crops with standard augmentations. **Mixed**: Teacher processes 2 global crops; student processes 2 global crops, 3 semantically masked global views (Masked G1-G3), 3 standard local crops, and 3 masked local crops (Masked L1-L3), combining both augmentation types. **Masked**: Teacher processes original unaugmented image; student processes 3 semantically masked global views and 6 local crops extracted from masked images.

starts from 0.9, and linearly onward to 1 over 300K iterations. For patch-token loss, where a random token masking is used, the masking ratio is set to 0.4. The Ko-Leo loss weight is set to 0.1, and patch-token loss ($\mathcal{L}_{iBOT}$) is equally weighted to the class token loss ($\mathcal{L}_{DINO}$). We implement the canonical multi-crop augmentation strategy from the official DINO repositories, which consists of two categories: global views with lighter augmentations seen by both teacher and student networks, and local crops with stronger augmentations seen only by the student network. Both the teacher and student networks process two global crops at $224 \times 224$ resolution, augmented using random resized crop with a scale range of $(0.4, 1.0)$, random horizontal flipping ($p = 0.5$), color jittering with brightness and contrast adjustments of $\pm 0.4$, saturation $\pm 0.2$, and hue $\pm 0.1$ (applied with probability 0.8), random grayscale conversion ($p = 0.01$), and Gaussian blur with kernel size 3 and sigma range $(0.1, 0.15)$, with the second global view additionally applying random solarization with threshold 64 ($p = 0.5$). The student network additionally processes 8-10 local crops at $96 \times 96$ resolution, with a more aggressive scale range of $(0.05, 0.4)$. For the semantic masking variant, we replace the 8-10 random local crops with 6 local crops (2 per masked image from our 3 generated masks), maintaining the same $96 \times 96$ resolution and augmentation pipeline, while the student additionally processes the three full-frame semantically masked images at $224 \times 224$ resolution, with the teacher continuing to see only the two unmasked global views.

**Training Data:** We utilize histopathological images from The Cancer Genome Atlas (TCGA) The Cancer Genome Atlas Research Network et al. (2013); Liu et al. (2018), a comprehensive cancer genomics program containing H&E stained WSIs spanning 33 cancer types. For our training dataset, we extract $448 \times 448$ pixel tiles at 0.25 microns per pixel resolution (corresponding to $40\times$ magnifi-

cation). We sample approximately 6,000 such tiles per whole-slide image after Otsu thresholding for patches that contain tissue samples, resulting in a training dataset of approximately 55 million image tiles. During the training process, the extracted tiles are resized to 224×224 pixels, therefore corresponding to 20× magnification (0.5 $\mu$m per pixel) as prior works Campanella et al. (2025) have shown that this resolution is suited for clinically relevant tasks like mutation prediction in cancer histopathology.

**Benchmarking:** We perform benchmarking on three types of tasks: patch-level classification tasks, patch-level nuclei segmentation tasks, and slide-level mutation classification tasks. For patch-level classification tasks, we utilize five established datasets: MHIST Wei et al. (2021), PatchCamelyon (PCam) Veeling et al. (2018), BRACS Brancati et al. (2022), and MiDOG Aubreville et al. (2023) (adapted into a classification task). For patch-level nuclei instance segmentation tasks, we employ the PanNuke dataset Gamper et al. (2019; 2020). For slide-level classification tasks, we utilize two mutation prediction tasks from an in-house dataset for the following mutations EGFR (Epidermal Growth Factor Receptor) in Lung Adenocarcinoma (LUAD), and FGFR3 (Fibroblast Growth Factor Receptor 3) in Bladder Cancer (BLCA). Details of all datasets are provided in App. B.

## 4 RESULTS AND ANALYSIS

Table 2: **Performance comparison of training strategies across patch-level and slide-level benchmarks.** Results include only the final evaluated checkpoint. Results show mean ± 95% CI on test sets. Best results for each benchmark are shown in **bold**.

| Model | Patch Classification (AUC) | | | | Patch Segmentation | Slide Mutation (AUC) | |
|---|---|---|---|---|---|---|---|
| | PCam | MiDOG | MHIST | BRACS | PanNuke (AJI) | LUAD-EGFR | BLCA-FGFR3 |
| Standard | $0.962 \pm 0.002$ | $0.675 \pm 0.023$ | $0.855 \pm 0.036$ | $0.908 \pm 0.009$ | $\textbf{0.615} \pm \textbf{0.005}$ | $0.523 \pm 0.033$ | $0.725 \pm 0.037$ |
| Masked | $0.944 \pm 0.006$ | $0.633 \pm 0.040$ | $0.837 \pm 0.021$ | $0.890 \pm 0.017$ | $0.608 \pm 0.005$ | $0.507 \pm 0.023$ | $0.646 \pm 0.023$ |
| Mixed | $\textbf{0.965} \pm \textbf{0.004}$ | $\textbf{0.675} \pm \textbf{0.021}$ | $\textbf{0.869} \pm \textbf{0.026}$ | $\textbf{0.921} \pm \textbf{0.011}$ | $0.609 \pm 0.005$ | $\textbf{0.577} \pm \textbf{0.028}$ | $\textbf{0.758} \pm \textbf{0.020}$ |

**Benchmark performance:** Figures 3 and 4a present patch-level and slide-level benchmark results across training iterations, showing means with 95% confidence intervals. Detailed methodology is provided in Appendix B, and the consolidated final evaluated checkpoint results are compared in table 2. Slide-level evaluation was limited to five intermediate checkpoints due to computational constraints from processing large numbers of WSIs. The semantic masking-only variant consistently underperforms on both patch and slide-level classification tasks, though it unexpectedly achieves competitive performance on the PanNuke segmentation task. The mixed augmentation strategy demonstrates consistent gains, particularly for median performance across training iterations. It also achieves superior median performance across all classification tasks at both patch and slide levels, indicating that semantic masking augmentation integrates effectively with state-of-the-art self-supervised learning frameworks when combined with standard augmentations rather than used in isolation.

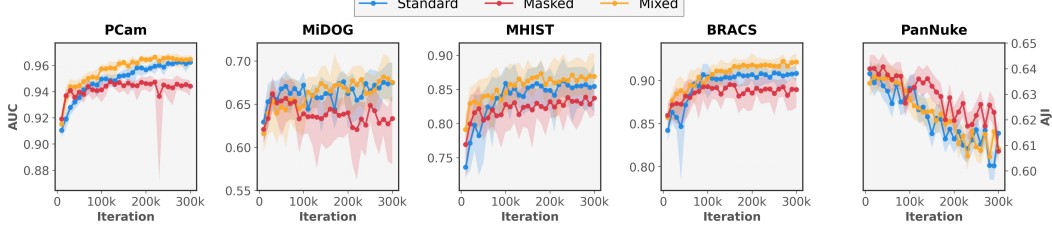

Figure 3: **Patch-level benchmark performance across training iterations.** Classification tasks (PCam, MiDOG, MHIST, BRACS) measured by AUC; segmentation task (PanNuke) measured by AJI Kumar et al. (2017). Error envelopes represent 95% confidence intervals from Monte Carlo cross-validation (classification) or independent training runs (segmentation).

**Ablations:** We conducted two key ablations to validate our approach. (1) Augmentation strategy ablation: Comparing Standard, Mixed, and Masked-only strategies. (2) Framework ablation: Com-

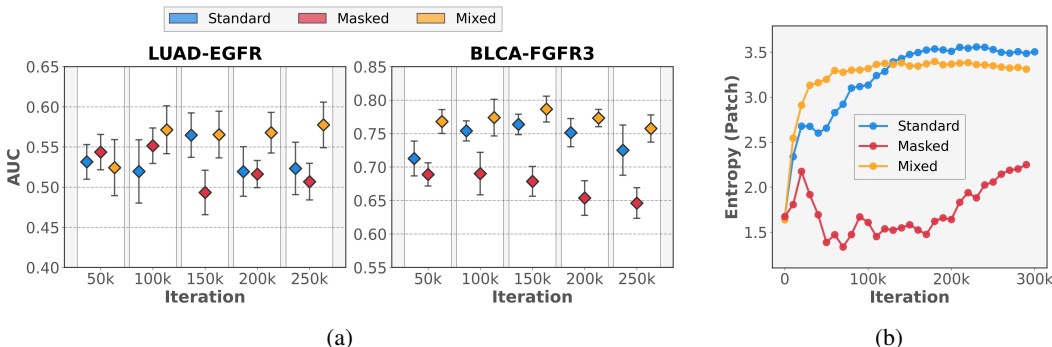

(a)                                        (b)

Figure 4: **Slide-level task performance and patch-token representation quality of DINOv2 training variants.** (a) Slide-level mutation prediction showing 10-fold cross-validation AUC scores for EGFR (LUAD) and FGFR3 (BLCA) mutations across five training iterations. Means (diamonds) and 95% confidence intervals are shown. Horizontal offsets within iterations provide visual clarity only. (b) Patch token Gram matrix entropy measuring semantic clustering evolution during training. Higher entropy indicates more diverse patch representations. Both panels compare Standard, Masked, and Mixed training strategies.

paring DINOv1 vs. DINOv2 implementations. These are detailed in Appendix B.5. These ablations confirm that the benefits of semantic masking are most pronounced when integrated with the more advanced DINOv2 framework.

**Patch token statistics:** We compute the entropy of patch token Gram matrices to assess self-similarity clustering, as $224 \times 224$ histopathology images at $20 \times$ magnification contain repetitive structures (e.g., nuclei, stroma, and epithelium). The calculation approach has been described in C and the results are shown in figure 4b. Notably, masked-only variants exhibit lower patch Gram matrix entropy, indirectly indicating fewer distinct clusters in the self-similarity space. This specialization may explain their competitive median performance on PanNuke segmentation as seen in figure 3, i.e., the reduced cluster diversity could benefit nuclei instance segmentation task specifically. An alternative observation could be that the low rank patch Gram matrix structure, while beneficial for dense tasks of the nuclei instance segmentation type, might be detrimental for classification tasks which require a better conceptual separation of the various tissue classes present in an image. This aspect is better explored in patch-token PCA maps, which is subsequently discussed.

**Patch token PCA visualizations:** Principal component analysis of patch tokens reveals distinct tissue decomposition patterns across training variants (methodology in Appendix D). Figure 5 shows representative gastric adenocarcinoma tissue containing smooth muscle bundles with adjacent cellularity and malignant glandular structures. The hue spectra distributions provide quantitative assessment of feature separation, where well-separated lobes, only when viewed in conjunction with the PCA plots, indicate successful decomposition of tissue microenvironment components. Despite exhibiting distinct spectral peaks, masked-only variants demonstrate poor spatial differentiation in PCA projections. This limited discriminative capability correlates with inferior classification performance, suggesting inadequate global feature learning. In contrast, mixing both masked and standard augmentations achieves superior component separation, distinguishing general cellularity (green channel) from blood cells in vascular regions (lower quadrant inner boundaries). The combination of well-separated hue spectra and clear spatial differentiation indicates enhanced tissue microenvironment representation in the mixed variant, providing insight into its superior patch and slide-level classification performance.

## 5 CONCLUSIONS

**Summary:** This work establishes that incorporating domain-specific inductive biases through semantic masking enhances representation quality in modern self-supervised frameworks without architectural modifications. When combined with standard augmentations in a mixed strategy, semantic masking yields consistent improvements across patch-level and slide-level tasks. The learned

masks successfully capture biologically meaningful components without supervision, as validated through PCA visualizations showing superior feature separation in the mixed augmentations variant. Our findings reveal that augmentation strategies have task-specific effects: while masked-only variants underperform on classification, they achieve competitive nuclei segmentation results. This was highlighted in the low-rank structure of the patch token Gram matrix produced using this augmentation type, indicating specialization, paving way for future work.

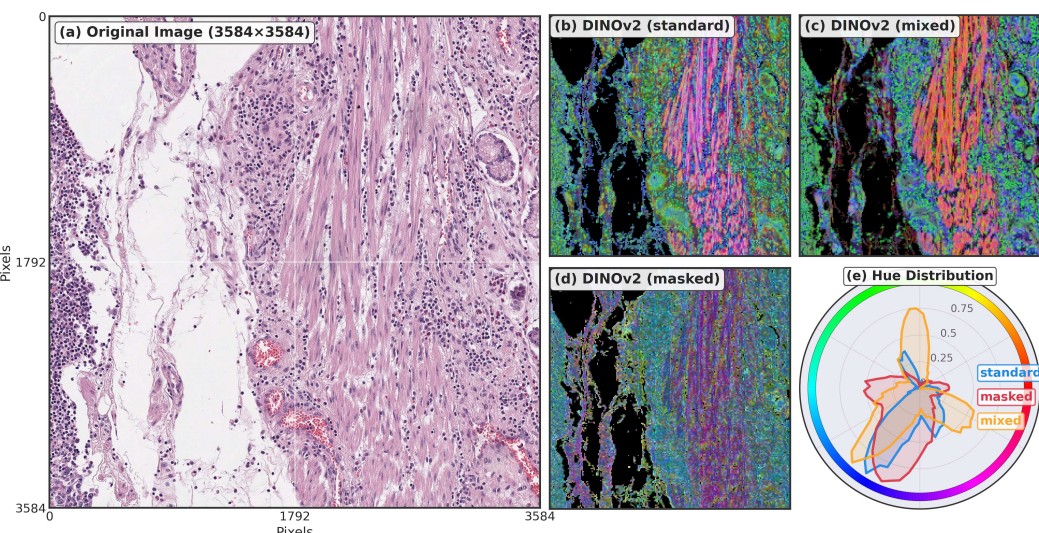

Figure 5: **Principal component analysis of patch tokens on a section of Gastric Adenocarcinoma (STAD) from the TCGA-STAD dataset.** The Cancer Genome Atlas Research Network et al. (2013) slide id: TCGA-D7-A6ET-01Z-00-DX1.A4FF5141-6B2A-456B-9EA2-E5DE72156647. PCA plots for the DINOv1 runs in the ablation study, provided alongside pathologist-level discussions, can be found in D.2.

**Limitations.** We focused this study on a ViT-B backbone Dosovitskiy et al. (2020) trained on ∼55M histopathology tiles, prioritizing rigorous downstream evaluation across diverse benchmarks. Compute constraints limited our ability to scale all architectures, but this controlled scope allowed us to isolate the contribution of semantic masking and demonstrate its impact consistently across patch- and slide-level tasks. While additional ablations—such as random masking or varying the number of masks $K$—could provide further confirmation, our results establish a benefit of incorporating semantic priors.

**Future work:** We will investigate optimal mask counts beyond K=3 and explore the unsupervised segmentation mask degradation in current ADIOS based approaches. Alternative masking strategies will be explored, including those that involve integration with vision-language models. Other blends of mixed-augmentations should be explored with approaches to include semantically guided patch-masking for the iBOT loss. Based on these results, we believe the mixed masking strategy will be competitive with or improve over standard DINOv2 pathology foundation models when trained with similar dataset sizes and ViT architectures.

**Reproducibility statement:** We ensure reproducibility through the use of open-source TCGA data for training and publicly available patch-level benchmark datasets. All experimental procedures, including model training, benchmarking protocols, and analysis methods for generating visualization plots, are extensively detailed in the main text and appendices. Training code will be made available as supplementary material upon acceptance. While slide-level mutation prediction benchmarks utilize proprietary institutional datasets and infrastructure for clinical validation, these experiments can be reproduced using publicly available WSI collections with alternative mutation targets. The specific mutations evaluated (EGFR in LUAD, FGFR3 in BLCA) represent standard biomarkers that could be substituted with other clinically relevant genomic alterations available in public repositories.

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

# A  DATA AUGMENTATION DETAILS

The following tables, tables 3, 4, and 5, specify the transformation operations applied during training.

Table 3: Teacher network augmentation pipeline

| Configuration | Input | Operations |
|---|---|---|
| Standard/Mixed | Global View 1 (224×224) | RandomResizedCrop (224×224, scale=[0.4, 1.0]) RandomHorizontalFlip ($p$=0.5) ColorJitter (brightness=0.4, contrast=0.4, saturation=0.2, hue=0.1, $p$=0.8) RandomGrayscale ($p$=0.01) GaussianBlur (kernel=3, $\sigma$=[0.1, 0.15]) Normalize (mean=[0.682, 0.564, 0.723], std=[0.162, 0.171, 0.139]) |
| | Global View 2 (224×224) | Same as Global View 1, plus: RandomSolarize (threshold=64, $p$=0.5) |
| Masked | Original Image (224×224) | None (normalization only) |

Table 4: Student network global views (224×224)

| Configuration | View Type | Operations |
|---|---|---|
| Standard | Global 1-2 | Same as teacher views |
| Mixed | Global 1-2 Masked Global 1-3 | Same as teacher views Semantic mask via frozen ADIOS (iter 40k) $x_{\text{masked}} = x \odot (1 - m_k), m_k \in [0, 1]^{H \times W}$ |
| Masked | Masked Global 1-3 | Semantic masking only |

Table 5: Student network local crops (96×96)

| Configuration | View Type | Operations |
|---|---|---|
| Standard | Local 1-8 | RandomResizedCrop (96×96, scale=[0.05, 0.4]) RandomHorizontalFlip ($p$=0.5) ColorJitter (brightness=0.4, contrast=0.4, saturation=0.2, hue=0.1, $p$=0.8) RandomGrayscale ($p$=0.01) GaussianBlur (kernel=3, $\sigma$=[0.1, 0.15]) Normalize (mean=[0.682, 0.564, 0.723], std=[0.162, 0.171, 0.139]) |
| Mixed | Standard Local 1-3 Masked Local 1-3 | Same as Standard configuration RandomCrop (96×96) from masked globals Standard local augmentations applied |
| Masked | Masked Local 1-6 | RandomCrop (96×96) from masked globals 2 crops per masked image, no additional augmentation |

# B BENCHMARKING

## B.1 PATCH LEVEL DATASETS

We evaluate our method on five histopathology datasets spanning classification and segmentation tasks.

**PCam** Veeling et al. (2018): A lymph node dataset collection of 327,680 $96 \times 96$ tiles extracted from histopathologic slide scans, along with a binary label indicating presence of metastatic tissue. The dataset contains 262,144 training and 65,536 test patches.

**MiDOG** Aubreville et al. (2023): A mitosis detection dataset containing 24,819 $96 \times 96$ patches from multiple tumor types and scanner vendors. The dataset originally contained bounding box targets for regression fitting, but this was adapted into a classification task for simplicity. Binary classification distinguishes mitotic figures from hard negatives, with 19,768 training and 5,051 test patches.

**MHIST** Wei et al. (2021): A colorectal polyp classification dataset containing 3,152 images of size $224 \times 224$ pixels, with binary labels of hyperplastic polyp (benign) or sessile serrated adenoma (precursor).

**BRACS** Brancati et al. (2022): A breast carcinoma subtyping dataset containing 4,221 regions of interest at $3173 \times 3345$ resolution from whole slide images. The dataset includes 7 classes spanning benign and malignant lesions, with 3,653 training and 568 test images. Due to the large native resolution ($3173 \times 3345$), BRACS images require resizing to match standard input dimensions for pretrained models.

**PanNuke** Gamper et al. (2020): A large multi-organ nuclei instance segmentation dataset containing 189,744 nuclei instances across 19 tissue types. Images are $224 \times 224$ pixels at $40\times$ magnification. The official 3-fold cross-validation splits are used, with folds 1-2 for training/validation (5,179 patches) and fold 3 for testing (2,722 patches).

## B.2 SLIDE LEVEL DATASETS

We analyzed two cohorts. In lung adenocarcinoma (n=923), we restricted genomic features to EGFR mutation status: 273 EGFR-positive cases (29.6%) and 650 EGFR-negative cases (70.4%). Specimens comprised 578 primary tumors (62.6%) and 345 metastases (37.4%). In bladder urothelial carcinoma (n=2,032), we used FGFR3 mutation status only: 393 FGFR3-positive cases (19.3%) and 1,639 FGFR3-negative cases (80.7%). The bladder specimens comprised 1,641 primary tumors (80.8%) and 391 metastases (19.2%). These binary mutation indicators were the only genomic inputs used in downstream analyses.

Genomic profiling was performed in-house at a CLIA-certified, hybrid-capture next-generation sequencing assay Zehir et al. (2017). DNA was extracted from formalin-fixed, paraffin-embedded tumor tissue (with matched peripheral blood when available), libraries underwent hybrid capture with custom baits targeting exonic regions and selected introns of up to 503 clinically relevant cancer genes, and high-depth sequencing was performed on Illumina instruments. A validated bioinformatics pipeline identified somatic single-nucleotide variants and small insertions/deletions, copy-number alterations, and selected structural rearrangements; the availability of matched normal enabled germline filtering.

## B.3 DATA PROCESSING PIPELINE

During benchmarking, datasets undergo the following preprocessing:

**Classification datasets**: Images are loaded as RGB and any resize transforms are applied if specified during dataset initialization. The specific preprocessing depends on the benchmarking configuration.

**Segmentation datasets**: During training, spatial transformations (rotations, flips) are synchronized between images and masks to maintain correspondence. Image-only augmentations (ColorJitter, blur, downscaling) can be applied based on configuration. Instance masks are converted to binary nuclei/background channels, and horizontal/vertical distance maps from nuclei centroids are

computed for improved boundary delineation. All transformations use configurable probabilities determined by the transform settings.

**Whole slide images**: WSIs undergo multi-scale processing for mutation prediction tasks. Thumbnails are generated at 0.5 microns per pixel resolution for tissue detection via Otsu thresholding using scikit-image's threshold_otsu function on Gaussian-blurred grayscale images. Additional filtering removes regions with low color variance (standard deviation $< 5$) to exclude non-tissue areas and marker artifacts. Following tissue detection, 448×448 pixel tiles are extracted with no overlap, comprehensively covering all tumor regions. Each tile is resized to 224×224 pixels and normalized using model-specific parameters before feature extraction.

**Normalization**: Images were normalized using the following statistics (mean=[0.6816, 0.5640, 0.7232], std=[0.1617, 0.1714, 0.1389]).

### B.4 BENCHMARKING PROCEDURE

#### B.4.1 CLASSIFICATION BENCHMARKS

We evaluate classification performance using frozen feature extraction followed by linear probing with Monte Carlo cross-validation for statistical robustness.

**Feature Extraction:** Features are extracted once from each dataset using the frozen backbone encoder $f_\theta$. For an input image $x_i$, we obtain $\mathbf{z}_i = f_\theta(x_i) \in \mathbb{R}^{768}$. All features are cached to disk to avoid redundant computation across multiple evaluation runs.

**Monte Carlo Cross-Validation:** We perform $N = 10$ iterations of stratified random splits to obtain robust performance estimates. Each iteration creates class-balanced 80/20 train/test splits, ensuring all classes are represented proportionally in both sets. The training set is further split 80/20 to create a validation set for hyperparameter tuning. For each iteration, we evaluate weight decay values $\lambda \in \{10^{-5}, 10^{-4}, 10^{-3}, 10^{-2}, 10^{-1}\}$ using a linear classifier. Training uses AdamW Loshchilov & Hutter (2017) optimization with learning rate $\eta = 0.1$ and cross-entropy loss, with early stopping triggered after 10 epochs without validation accuracy improvement. The model achieving optimal validation accuracy determines the best weight decay for that iteration.

**Evaluation Metrics:** The model with optimal hyperparameters is evaluated on the held-out test set. For binary classification, AUC is calculated using the positive class probabilities. For multi-class problems, we employ one-vs-rest (OvR) strategy with weighted averaging, where each class is treated as positive against all others and the resulting AUC scores are averaged weighted by class support. Final metrics are reported as mean $\pm$ standard deviation across all Monte Carlo iterations. We compute 95% confidence intervals from the empirical distribution of metrics, providing uncertainty estimates for model performance.

#### B.4.2 SEGMENTATION BENCHMARKS

Segmentation evaluation employs a CellViT Hörst et al. (2024) decoder trained on frozen backbone features for nuclei instance segmentation.

**Architecture:** CellViT combines the frozen encoder $f_\theta$ with a U-Net style decoder incorporating skip connections at multiple scales. The decoder produces three outputs: a binary segmentation mask $M \in [0,1]^{H \times W \times 2}$ distinguishing nuclei from background, and horizontal/vertical distance maps $D \in \mathbb{R}^{H \times W \times 2}$ encoding distances to nuclei centers for instance separation. Nuclei classification is avoided in this work.

**Loss function:** We optimize a combined loss function adapted from Hörst et al. (2024) (incl. terminology) $\mathcal{L} = \mathcal{L}_{\text{CE}} + \mathcal{L}_{\text{Dice}} + \alpha \mathcal{L}_{\text{MSE}} + \beta \mathcal{L}_{\text{MSGE}}$ with weights $\alpha = 2.5$, and $\beta = 8.0$. The cross-entropy loss $\mathcal{L}_{\text{CE}}$ handles pixel classification, while the Dice coefficient loss $\mathcal{L}_{\text{Dice}}$ with class weights [0.7, 0.3] improves segmentation overlap. The mean squared error $\mathcal{L}_{\text{MSE}}$ supervises distance map regression, and the mean squared gradient error $\mathcal{L}_{\text{MSGE}}$ preserves edge information by penalizing gradient differences between predicted and ground truth distance maps.

**Training:** The training data is split 80/20 for training and validation. We employ AdamW optimization with a warmup-decay learning rate schedule: warming up from $10^{-6}$ to $10^{-4}$ over 5 epochs, then linearly decaying to $10^{-5}$ over the remaining epochs. Gradient clipping at norm 0.3 ensures

training stability. Early stopping monitors validation AJI with patience of 10 epochs, and training runs for a maximum of 50 epochs with batch size 16. The encoder remains frozen throughout training, with only the decoder parameters updated.

**Instance Segmentation:** Predictions undergo post-processing to generate instance segmentation maps. The binary mask is thresholded at 0.95, followed by morphological operations including erosion and dilation to clean up predictions. Gradient maps from the horizontal and vertical distance predictions are computed using Sobel filters, normalized, and thresholded to identify nuclei boundaries. Markers for individual nuclei are generated from regions with high confidence and refined using morphological opening. The watershed algorithm is then applied using the distance transform as the topographic surface and the markers as seeds, producing the final instance segmentation.

**Evaluation Metric:** Performance is measured using the Aggregated Jaccard Index (AJI):

$$\text{AJI} = \frac{\sum_{i=1}^{N_{\text{GT}}} |G_i \cap P_{\sigma(i)}|}{\sum_{i=1}^{N_{\text{GT}}} |G_i \cup P_{\sigma(i)}| + \sum_{k \in U} |P_k|}, \tag{5}$$

where $G_i$ represents ground truth nuclei instances, $P_j$ represents predicted instances, $\sigma$ denotes the optimal bipartite matching between ground truth and predictions, and $U$ contains unmatched predictions. AJI jointly penalizes both segmentation errors and detection mistakes, providing a comprehensive measure of instance segmentation quality. We report mean, standard deviation, median, quartiles, and 95% confidence intervals of AJI scores across the test set.

### B.4.3 SLIDE-LEVEL MUTATION PREDICTION BENCHMARKS

We evaluate mutation prediction performance using multiple instance learning (MIL) with frozen feature extraction from WSIs.

**Feature Extraction and Aggregation:** WSIs are processed through the pipeline described in the data processing section. Each 224×224 pixel tile is passed through a pre-trained ViT-Base encoder $f_\theta$ to generate 768-dimensional feature embeddings. Five different model checkpoints are evaluated at iterations 50,000, 100,000, 150,000, 200,000, and 250,000 to assess the impact of training duration on downstream performance. Features from all tiles within a WSI are stored for subsequent MIL aggregation. Tile-level features are aggregated using a Gated Multi-head Attention (GMA) mechanism Ilse et al. (2018) to produce slide-level representations. Each checkpoint evaluation runs independently on high-memory NVIDIA GPUs (A100/H100 with 80GB memory) to accommodate the large number of tiles per WSI and parallel feature extraction across multiple slides.

**Binary Classification:** The aggregated slide representation $\mathbf{h}_{\text{slide}}$ is passed through a binary classifier for mutation prediction. Training employs AdamW optimization with learning rate $\eta = 0.001$, batch size 32, and weight decay $\lambda = 10^{-4}$ for 200 epochs. These hyperparameters were optimized specifically for the mutation prediction datasets.

**Cross-Validation:** Model evaluation uses 10-fold stratified cross-validation to ensure robust performance estimation and account for class imbalance in mutation labels. Each fold maintains the original positive/negative class ratio (29.6% EGFR-positive in LUAD, 19.3% FGFR3-positive in BLCA). Performance is reported as mean AUC $\pm$ 95% confidence intervals across all folds.

**Evaluation Metric:** AUC, in the same manner as the patch-level classification tasks.

### B.5 ABLATION STUDIES: FRAMEWORK AND AUGMENTATION ANALYSIS

This section presents comprehensive ablation studies examining two critical design choices in our approach: (1) the augmentation strategy (Standard, Mixed, or Masked-only) and (2) the self-supervised learning framework (DINOv1 vs. DINOv2). These ablations validate that the benefits of semantic masking are most pronounced when integrated with modern self-supervised frameworks and combined with standard augmentations. We trained models using identical hyperparameters across both frameworks, with the only differences being the framework-specific components. Table 6 shows the augmentation configurations for DINOv1, which parallel those used for DINOv2 (shown in Table 1 of the main text). All models were trained on the same 55 million TCGA tiles for 300K iterations, with evaluations performed at regular intervals. The mask generator checkpoint

Table 6: **Training configurations for augmentation strategies for ablation study.** Settings for global and local crops across Standard, Mixed, and Masked variants for DINOv1 and DINOv2. Standard uses conventional augmentations only; Mixed combines standard and semantic masking; Masked uses semantic masking exclusively.

| Method | Variant | Global Crops | | Local Crops | | Total Views |
|--------|---------|--------------|--------|-------------|--------|-------------|
| | | Standard | Masked | Standard | Masked | |
| DINOv2 | Standard | 2 | 0 | 8 | 0 | 10 |
| | Mixed | 2 | 3 | 3 | 3 | 11 |
| | Masked | 0 | 3 | 0 | 6 | 9 |
| DINOv1 | Standard | 2 | 0 | 10 | 0 | 12 |
| | Mixed | 2 | 3 | 3 | 3 | 11 |
| | Masked | 0 | 3 | 0 | 6 | 9 |

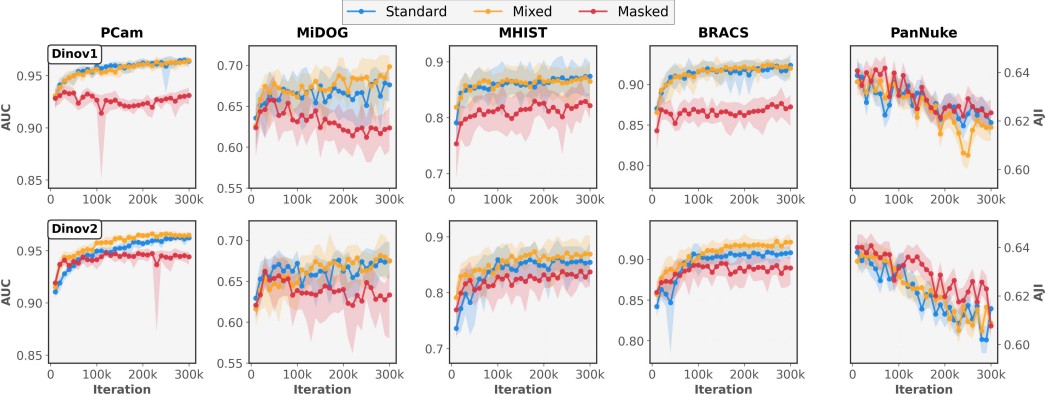

Figure 6: **Patch-level benchmark performance across training iterations.** Classification tasks (PCam, MiDOG, MHIST, BRACS) measured by AUC; segmentation task (PanNuke) measured by AJI Kumar et al. (2017). Top: DINOv1 variants; Bottom: DINOv2 variants. Error envelopes represent 95% confidence intervals from Monte Carlo cross-validation (classification) or independent training runs (segmentation).

from iteration 40,000 was used consistently across all experiments. The patch and slide level results are provided in figures 6 and 7. It is clear from the results that the advantage of the segmentation masks when used as a data augmentation is more pronounced in DINOv2 results over DINOv1 results.

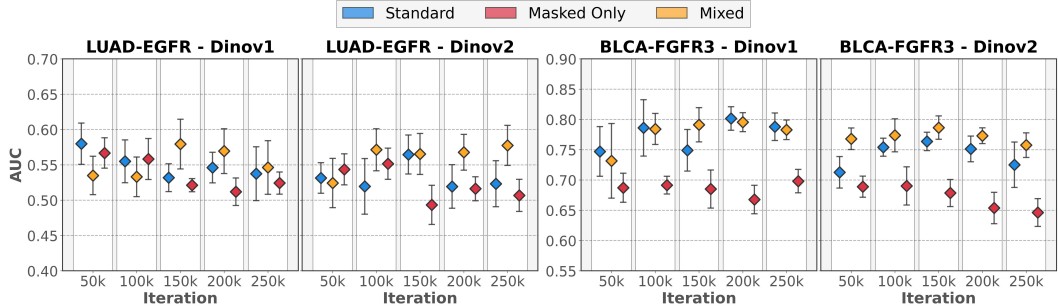

Figure 7: **Slide-level mutation prediction performance across five intermediate training iterations.** Strip plots showing 10-fold cross-validation AUC scores for EGFR (LUAD) and FGFR3 (BLCA) mutations. Means (diamonds), and 95% confidence intervals are shown. Horizontal offsets within iterations for visual clarity only; all points at each x-position faceted by the gray borders represent the same checkpoint iteration number for each variant.

## C  REPRESENTATION STATISTICS

Features are extracted from all classification datasets using the frozen encoder $f_\theta$ and concatenated into a feature matrix $\mathbf{F} \in \mathbb{R}^{N \times d}$ where $N$ is total samples and $d$ is feature dimension. An adaptive convergence framework incrementally samples features (starting at 10,000, adding 1,000 per iteration) with 100-iteration bootstrap validation until variance stabilizes below $10^{-3}$ for 5 consecutive steps.

**RankMe** Garrido et al. (2023) measures effective rank through eigenvalue entropy. After computing the covariance matrix $\hat{\mathbf{C}} = \frac{1}{N}\mathbf{F}^T\mathbf{F}$ and obtaining singular values $\{\sigma_i\}$:

$$\text{RankMe}(\mathbf{F}) = \exp\left(-\sum_i p_i \log p_i\right), \quad p_i = \frac{\sigma_i}{\sum_j \sigma_j} \tag{6}$$

Higher values indicate more diverse representations without dimensional collapse.

$\alpha$-**ReQ** Agrawal et al. (2022) fits a power-law $\lambda_k \propto k^{-\alpha}$ to the eigenspectrum, where $\lambda_k$ is the $k$-th eigenvalue, via least-squares in log-log space:

$$\log \lambda_k = -\alpha \log k + c \tag{7}$$

Models with decay coefficient $\alpha \in [1.0, 2.0]$ and $R^2 > 0.9$ demonstrate superior generalization.

**CLID** Lu et al. (2023) combines Cluster Learnability (CL) with Intrinsic Dimension (ID). CL measures $k_{nn}$-nearest neighbor accuracy ($k_{nn} = 5$) on K-means pseudo-labels using $k = 7$ clusters (average number of classes). ID is estimated via TwoNN method using distance ratios $\mu_i = r_{2,i}/r_{1,i}$ where $r_{1,i}, r_{2,i}$ are distances to first and second nearest neighbors:

$$\text{ID} = \left(\frac{1}{N}\sum_{i=1}^{N} \log \mu_i\right)^{-1}, \quad \text{CLID} = \text{CL} \times \left(1 - \exp\left(-\frac{\text{ID}}{d}\right)\right) \tag{8}$$

All metrics report bootstrap mean, standard deviation, and 95% confidence intervals at convergence.

**Patch Token Entropy:** Patch token clustering is evaluated through entropy of the Gram matrix singular value distribution. For each checkpoint, patch embeddings $\mathbf{P}_i \in \mathbb{R}^{196 \times 768}$ are extracted from 1600 images (100 batches of 16 images), excluding CLS and register tokens. After L2 normalization, the Gram matrix $\mathbf{G} = \mathbf{P}\mathbf{P}^T$ captures pairwise cosine similarities between patches. Singular values $\{\sigma_i\}$ are obtained via SVD and entropy is computed as:

$$H = -\sum_{i=1}^{196} p_i \log p_i, \quad \text{where} \quad p_i = \frac{\sigma_i}{\sum_j \sigma_j} \tag{9}$$

Lower entropy indicates stronger clustering of semantically similar patches. Results are reported at 10,000 iteration intervals with 95% bootstrap confidence intervals.

## D  PATCH TOKEN PCA VISUALIZATIONS

### D.1  PROCEDURE

For visualization, we extract 3584×3584 pixel regions at 0.25 $\mu$m/pixel from whole slide images. We bias our region selection to boundary regions of WSI, so that multiple region classes can be expected in the selection. Each region is divided into 256 non-overlapping $224 \times 224$ patches in a $16 \times 16$ grid. Features are extracted using trained encoders, producing 196 patch tokens ($14 \times 14$ spatial resolution) per patch, yielding feature matrix $\mathbf{F} \in \mathbb{R}^{50176 \times 768}$.

**Whitespace Separation:** Since white-space could be intra-tissue and thus relevant, relative to whitespace outside tissue regions, a PCA and consensus based background removal is applied so that the PCA is applied in the relevant region over the same number of pixels for all models trained. The final checkpoint at iteration 300k is used for the analysis. Features are reduced to 3 dimensions via PCA. K-means clustering with $k = 4$ identifies background, light tissue, medium tissue, and

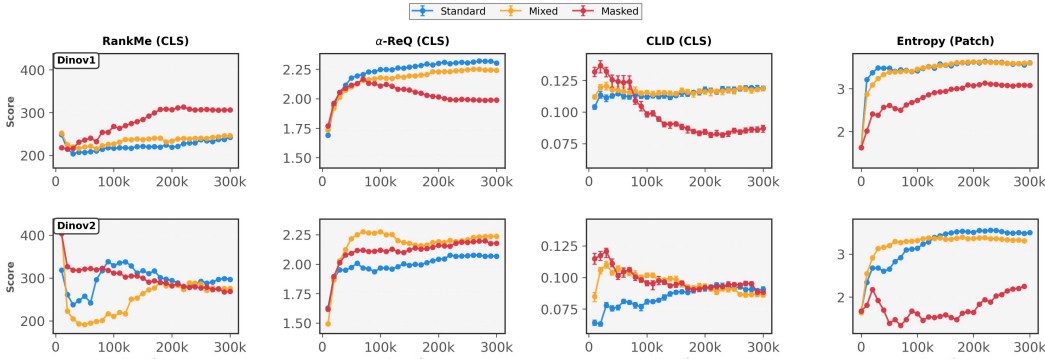

Figure 8: **Task-agnostic representation quality metrics across training iterations.** CLS token metrics: RankMe (effective rank), $\alpha$-ReQ (eigenspectrum decay), and CLID (cluster learnability $\times$ intrinsic dimension) computed from concatenated patch-level dataset features. Patch token metric: Entropy of Gram matrix singular value distribution measuring semantic clustering. Top: DINOv1 variants; Bottom: DINOv2 variants. Error bars show 95% bootstrap confidence intervals. For classification tasks, CLS token metrics fail to predict the relative performance differences observed across augmentation strategies. RankMe, $\alpha$-ReQ, and CLID show no consistent patterns distinguishing the superior mixed variant from the inferior masked-only approach. This disconnect suggests these metrics, developed primarily for natural images, may not capture the relevant structure in histopathology representations. The high non-linearity and domain-specific characteristics of pathology images, for example uniform staining, presence of large number of classes in a batch, the lack of perspective, and repetitive textures, likely require alternative quality measures that better reflect the hierarchical tissue organization critical for diagnostic tasks.

dark tissue clusters. Each cluster is scored for background likelihood based on: mean brightness ($>225$: +1 to +3 points), brightness uniformity (std $<10$: +1), peripheral location ($>50\%$ at edges: +1), spatial coherence ($\leq 2$ connected components: +1), and extreme PC1 values (+1). Clusters scoring $\geq 4$ are labeled as background. Morphological operations preserve tissue by removing small isolated background regions within patches. Validation metrics include whitespace recall and tissue preservation rates. When processing multiple models, individual foreground masks are combined via intersection to create a consensus mask. If the consensus contains $<1000$ pixels, union is used instead to ensure sufficient foreground pixels for analysis.

**PCA Calculation and Visualization:** PCA is applied to consensus foreground features only, projecting to 3 dimensions. Components are normalized to [0,1] and mapped to RGB channels. The resulting 196-dimensional vectors per patch are reshaped to $14 \times 14$ spatial grids and assembled into the final $224 \times 224$ visualization. Background pixels remain black. RGB values are converted to HSV color space. Hue distributions (0-360°) are computed and visualized as polar histograms with individual min-max normalization. A color ring at radius 1.12-1.20 provides hue reference.

### D.2 Histopathological description of PCA Samples

We present PCA visualizations from all training configurations at iteration 300,000, projecting learned representations into RGB space to assess semantic decomposition quality. The DINOv2-mixed augmentation configuration emerges as our best-performing model, exhibiting superior spectral separation in the hue distributions with distinct, non-overlapping colors corresponding to biologically meaningful tissue components. Based on this, we provide detailed histopathological annotations for the DINOv2-mixed PCA visualizations in the figure captions. Readers should compare these reference visualizations against other training configurations, noting both the polar histogram distributions (where tighter, more separated peaks indicate better feature disentanglement) and the correspondence between color components and tissue structures described in each caption. The histopathological interpretations are written at practitioner level, identifying specific architectural patterns, cellular morphologies, and microenvironmental features that trained pathologists recognize in routine diagnostic practice.

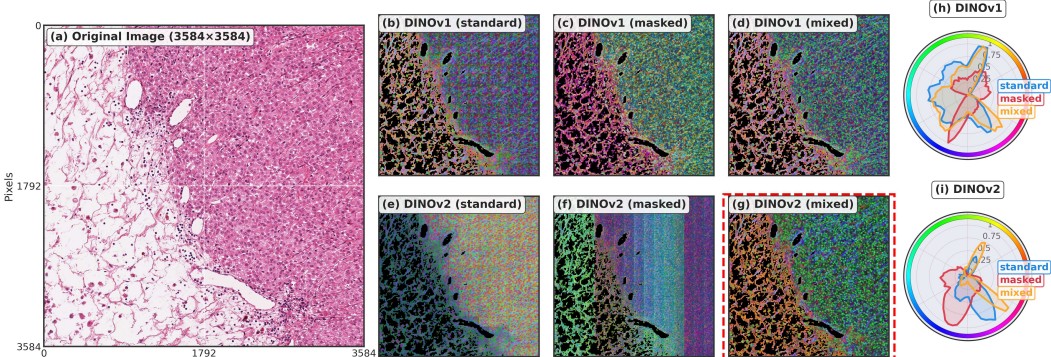

Figure 9: **TCGA-ACC, slide id: TCGA-OR-A5LR-01Z-00-DX4.0AF1F52B-222F-4D41-94A1-AA7D9CFBC70C.** Red dashed line indicates image for which descriptors are documenting. Adrenocortical carcinoma demonstrating distinct tissue component separation through PCA visualization. The green component identifies malignant cortical cells, while blue highlights trabecular architectural patterns characteristic of neuroendocrine differentiation. Red components correspond to regions of homogeneous cortical tissue and acellular areas, derived from the prominent zona glomerulosa visible in the left portion of the image. The left region shows evidence of cortical necrosis and structural disintegration, representing advanced tumor progression.

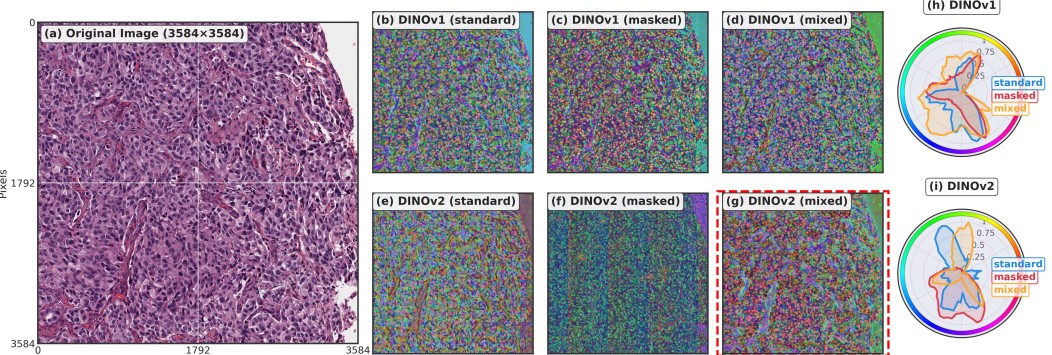

Figure 10: **TCGA-BLCA, slide id: TCGA-FJ-A3Z7-01Z-00-DX6.28B723F7-1035-4DC2-8DB1-87F08166A9FA.** Red dashed line indicates image for which descriptors are documenting. Bladder urothelial carcinoma with clear differentiation of microenvironmental components. The blue component delineates intratumoral vasculature, characterized by linear arrangements of endothelial cells and intravascular erythrocytes (appearing red in the original H&E stain). Orange components identify nests of malignant urothelial cells, while purple regions represent perivascular hemorrhage and extravasated blood. The green component captures low-density interstitial spaces separating vascular structures from tumor cell nests, demonstrating the model's ability to distinguish tissue microarchitecture.

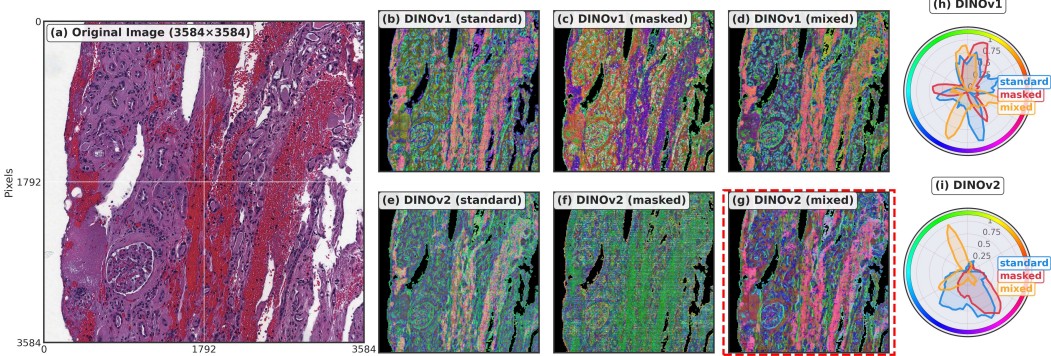

Figure 11: **TCGA-KIRC, slide id:  TCGA-DV-5576-01Z-00-DX1.ddd18b71-fc48-40f7-bc87-fb50d9ff468c.** Red dashed line indicates image for which descriptors are documenting. Clear cell renal cell carcinoma showing preserved renal architecture. The bottom left quadrant reveals a glomerular structure where blue components identify the cellular glomerular tuft, while green represents the urinary space and interstitial regions. Red components correspond to presence of blood, typical of kidney tissue samples. The upper panels display tubular structures with epithelial cells mapped to blue-brown components, indicating the model's discrimination between functionally distinct renal compartments despite similar cellular morphology.

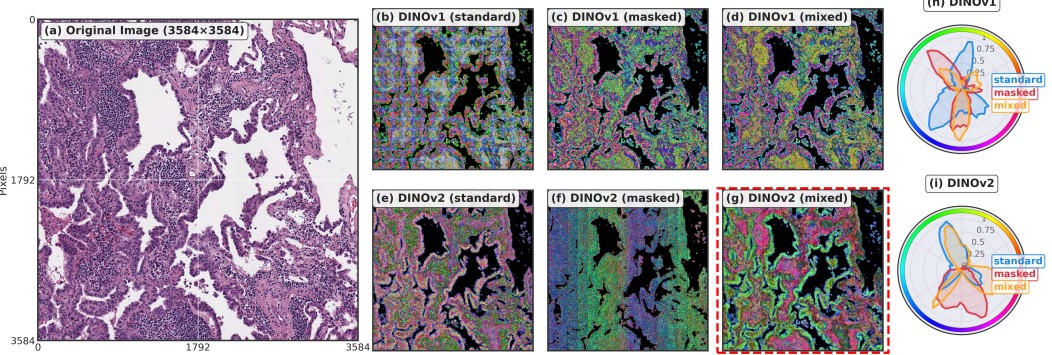

Figure 12: **TCGA-LUAD, slide id:  TCGA-MP-A4TH-01Z-00-DX1.E89D2C19-F9B2-4BF2-AA5F-6104CBC076D1.** Red dashed line indicates image for which descriptors are documenting. Lung adenocarcinoma exhibiting lepidic growth pattern. The green component precisely delineates malignant epithelial structures lining alveolar walls, a defining feature of lepidic-predominant adenocarcinoma. Red components identify lymphocytic infiltrates within the stromal compartment, indicating immune response. Blue regions correspond to collapsed residual alveolar spaces within the tumor mass, representing architectural distortion characteristic of invasive adenocarcinoma while maintaining spatial context of the original lung parenchyma.

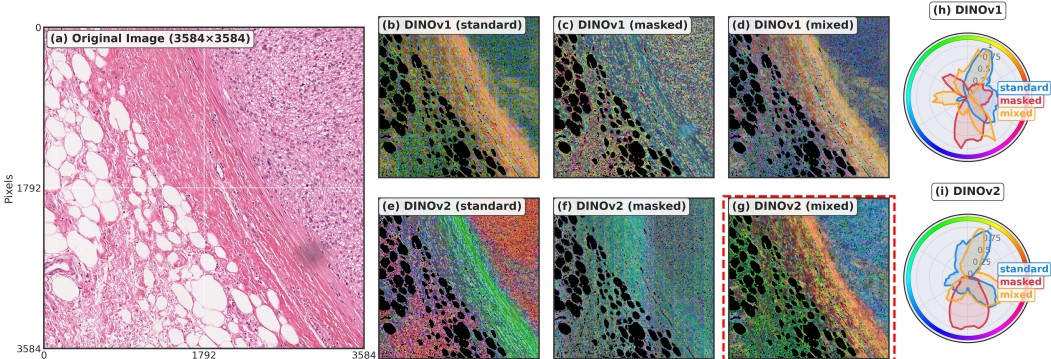

Figure 13: **TCGA-SARC, slide id: TCGA-IF-A4AJ-01Z-00-DX1.A6CE6AEC-B645-4885-A995-99FF7A4B26A5.** Red dashed line indicates image for which descriptors are documenting. Well-differentiated liposarcoma demonstrating adipocytic differentiation. Green components identify both mature adipocyte membranes in the left region and intracytoplasmic lipid vacuoles within liposarcoma cells, reflecting the tumor's adipocytic lineage. Blue components highlight the pleomorphic nuclei of malignant cells, exhibiting characteristic size and shape variability. Red-orange coloration delineates the fibrous tumor capsule and septae, providing structural compartmentalization typical of well-differentiated liposarcoma.

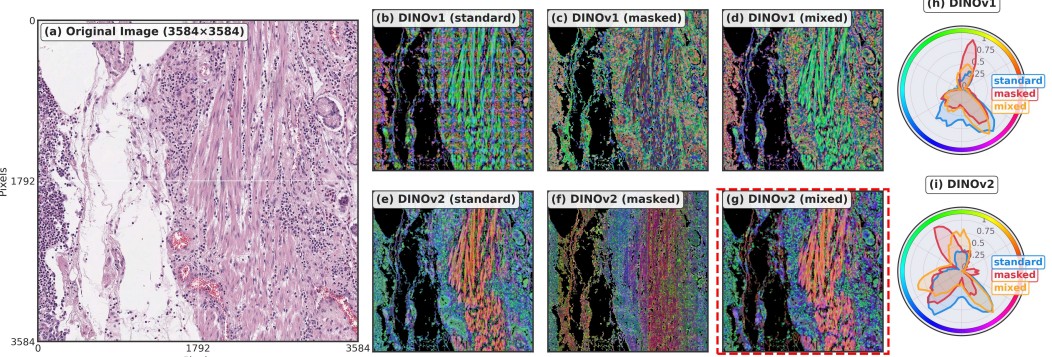

Figure 14: **TCGA-STAD, slide id: TCGA-D7-A6ET-01Z-00-DX1.A4FF5141-6B2A-456B-9EA2-E5DE72156647.** Red dashed line indicates image for which descriptors are documenting. Gastric adenocarcinoma with complex glandular architecture. Green components identify both lymphoid aggregates at tissue boundaries and epithelial cells within malignant glands (upper right panel), demonstrating the model's ability to recognize similar cellular densities in different contexts. Orange components trace smooth muscle bundles of the muscularis mucosae, while blue regions correspond to both mucin-producing cells within glandular lumina and myxoid stromal changes, reflecting the heterogeneous microenvironment characteristic of gastric adenocarcinoma.

# E  SEGMENTATION MAP EXAMPLES

This section provides qualitative evaluation of the semantic mask generation capabilities across training iterations. We extract 896×896 pixel regions from representative whole slide images from The Cancer Genome Atlas (TCGA) dataset for adrenocortical carcinoma (ACC), bladder urothelial carcinoma (BLCA), kidney renal clear cell carcinoma (KIRC), lung adenocarcinoma (LUAD), sarcoma (SARC), and stomach adenocarcinoma (STAD).

For each cancer type, we compare mask generators trained using the baseline ADIOS methodology against our proposed approach incorporating perceptual reconstruction loss. The mask models are evaluated at 10,000-iteration intervals (20k, 30k, 40k, 50k) to assess the evolution of semantic decomposition quality during training. Each 896×896 pixel region is processed through the frozen mask generator using non-overlapping 224×224 pixel tiles, producing three semantic masks per tile that are then assembled into the full visualization. Training the mask generator beyond a certain number of iterations causes deterioration of the mask quality, therefore in this work, iteration 40,000 was chosen for use in data-augmentation. The resulting segmentation maps demonstrate the model's ability to identify biologically meaningful tissue components without explicit supervision.

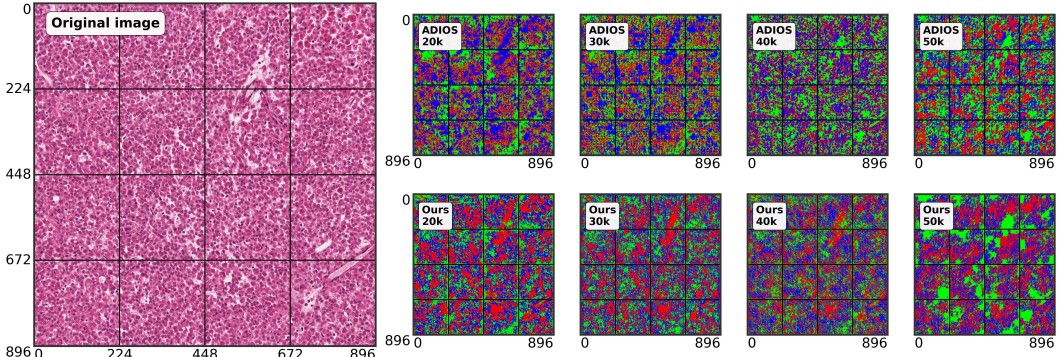

Figure 15: TCGA-ACC, slide id: TCGA-OR-A5LR-01Z-00-DX4.0AF1F52B-222F-4D41-94A1-AA7D9CFBC70C.

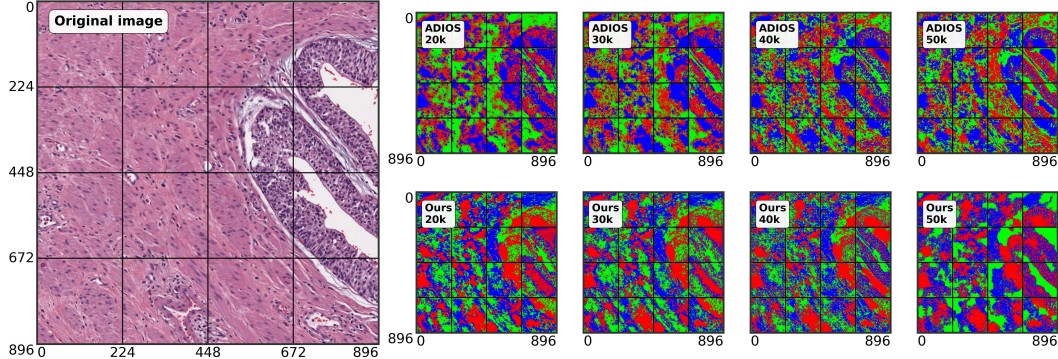

Figure 16: TCGA-BLCA, slide id: TCGA-FJ-A3Z7-01Z-00-DX6.28B723F7-1035-4DC2-8DB1-87F08166A9FA.

# F  LLM USAGE

Claude Sonnet and Opus versions 3.5, 4 and 4.1 were used for generating patch-level benchmarking and analysis code. Writing the draft was performed alongside Claude Opus 4.1, and ChatGPT 5 was utilized to produce a part of the slide-level mutation prediction code.

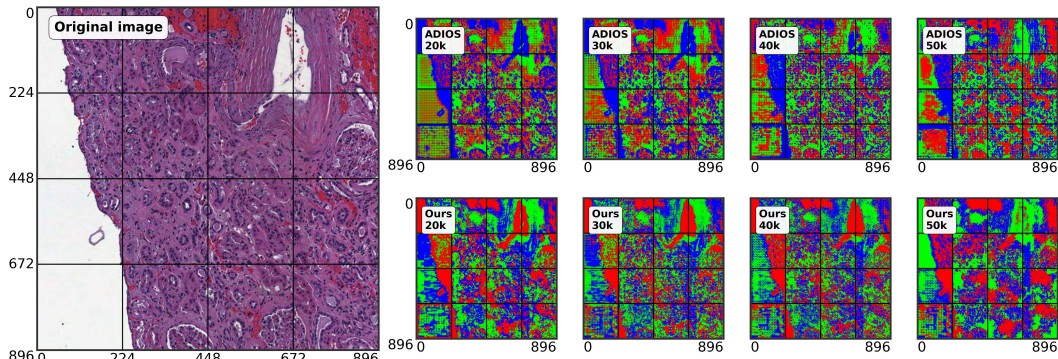

Figure 17: TCGA-KIRC, slide id: TCGA-DV-5576-01Z-00-DX1.ddd18b71-fc48-40f7-bc87-fb50d9ff468c.

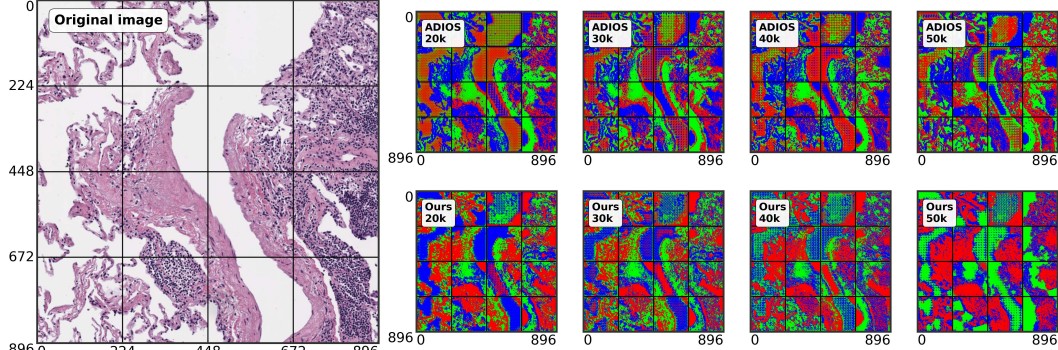

Figure 18: TCGA-LUAD, slide id: TCGA-MP-A4TH-01Z-00-DX1.E89D2C19-F9B2-4BF2-AA5F-6104CBC076D1.

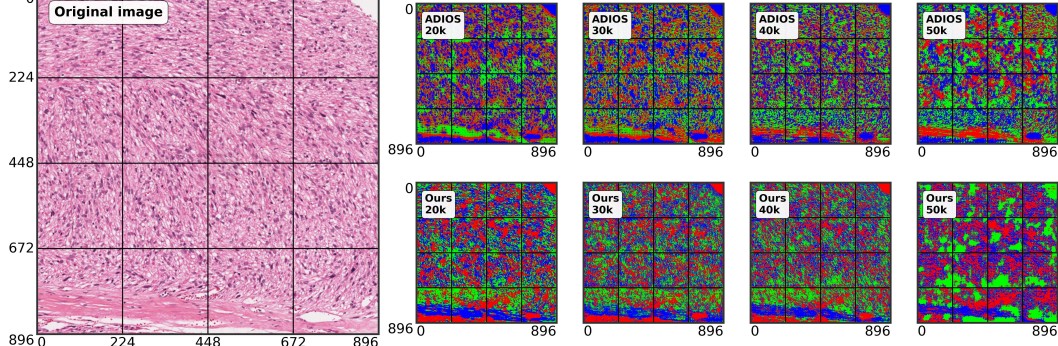

Figure 19: TCGA-SARC, slide id: TCGA-IF-A4AJ-01Z-00-DX1.A6CE6AEC-B645-4885-A995-99FF7A4B26A5.

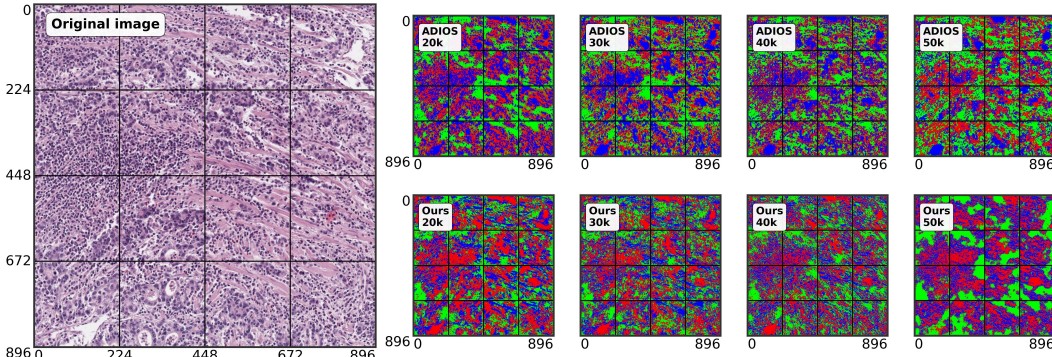

Figure 20: TCGA-STAD, slide id: TCGA-D7-A6ET-01Z-00-DX1.A4FF5141-6B2A-456B-9EA2-E5DE72156647.

