# OpenReview forum: "Tissue Microenvironment as an Additional Prior for Visual Representation Learning in Histopathology"
_ICLR.cc/2026/Conference — ICLR 2026 Conference Withdrawn Submission_

### Official Review · Reviewer_kdpz · 2025-10-27

**Soundness:** 3
**Presentation:** 3
**Contribution:** 2
**Rating:** 2
**Confidence:** 4

**Summary:**

The paper describes a self-supervised method for the analysis of histopathology image, exploiting DinoV2 for representation learning at patch level, and considering augmentation at tissue level with microenvironment information as additional information for improving the expressivity of latent representation. Authors assesses the proposed framework against ADIOS baseline and compare three different augmentation strategies including or excluding semantic masking. They also test their method in various downstream tasks.

**Strengths:**

1. The manuscript is well written and organized. The work is well motivated, and narrative flow makes the manuscript enjoyable and easy to read. All in all, I liked reading it.
2. The methodology is sound and most of technical details for reproduction are provided: anyway, the availability of source code would be preferrable.
3. Assessment is extensive, as well as comparison with respect to baselines, and the results appear convincing ans showcasing significant improvement.

**Weaknesses:**

All in all, the paper appears solid, but I am not sure it does provide significant contributions to the field, especially for what concerns learning representations.
Specifically:
1. Positioning with respect to SOTA. The critical discussion of related work appears limited, and I would expect a clear enumeration in that section of what are the main differences with respect to previous methods, and especially ADIOS. Moreover, most recent self supervised learning methods in histopathology (2025) are excluded from discussion, like for example
Sun, Y., Si, Y., Zhu, C., Gong, X., Zhang, K., Chen, P., ... & Yang, L. (2025). Cpath-omni: A unified multimodal foundation model for patch and whole slide image analysis in computational pathology. In Proceedings of the Computer Vision and Pattern Recognition Conference (pp. 10360-10371).
Han, M., Qu, L., Yang, D., Zhang, X., Wang, X., & Zhang, L. (2025). Mscpt: Few-shot whole slide image classification with multi-scale and context-focused prompt tuning. IEEE Transactions on Medical Imaging.
Xiang, Jinxi, Xiyue Wang, Xiaoming Zhang, Yinghua Xi, Feyisope Eweje, Yijiang Chen, Yuchen Li et al. "A vision–language foundation model for precision oncology." Nature 638, no. 8051 (2025): 769-778. (MUSK)
2. Incremental improvement. The idea of using tissue microenvironment prior and semantic masking provides some improvement in the pipeline, but it does not seem a breakthrough innovation in representation learning.
3. Assessment limitations. I would have expected that authors compare their outcomes with available and well established foundation models, to see what are the differences in terms of performance on downstream tasks;  I would at least consider MUSK for assessment.
4. Explainability and visualization. It is not clear to me how authors manage to represent visually PCA projection in slides; for all figures, I would expect that they provide insets to showcase details about the representation. How did you decide the colormapping schemes according to hue ? Please define a rationale, since visual explainability related to reported results does not appear clear.

**Questions:**

1. Improve visualization of qualitative results by: a) trying different colormapping strategies; b) adding detail insets to highlight the representation differences.
2. Update the discussion of related work: all recent foundation methods presented in 2025 are excluded from discussion.
3. Include additional and recent baselines in the comparison for downstream tasks, like done for example in MUSK paper.

**Details Of Ethics Concerns:**

No concerns

---

### Official Review · Reviewer_cCwz · 2025-10-31

**Soundness:** 2
**Presentation:** 2
**Contribution:** 2
**Rating:** 2
**Confidence:** 4

**Summary:**

Self-supervised learning for Digital Pathology relies on image augmentations inherited from previous work on natural images. This paper proposes instead to perform augmentations of images guided by information about the tissue micro-environment. In particular, they present a method to segment tissue tiles in an unsupervised fashion, leading to a set of semantic masks. The latter are used to generate global and local masked views fed to the student network in DINOv2-style pre-training. By doing so, the authors claim that the trained model will better learn to extract information about the biological structure displayed on H&E patches. The pre-trained model is evaluated on a set of patch-level and slide-level datasets.

**Strengths:**

- S1: Augmenting self-supervised learning for Digital Pathology with biologically-grounded priors is a relevant direction.
- S2: The contributions are clearly presented and motivated in the paper.
- S3: Experiments to validate the method are performed on both patch-level and slide-level datasets, which is appreciated. Curves showing the evolution of performance as a function of the number of epochs are also interesting.

**Weaknesses:**

- W1: [Major] The quality of the performed semantic mask generation is hard to assess from the paper. Indeed, only Figure 1 shows a qualitative example where we observe the input tile along with results from ADIOS and the proposed model. It is hard to compare the performance of their method compared with ADIOS, and also the absolute quality of the maps generated by the method itself. A quantitative study should be performed to better highlight the gain from their approach.

- W2: [Major] Related to W1, one could also wonder why using a trained semantic segmentation model such as Cellvit (Horst et al., Cellvit: Vision transformers for precise cell segmentation and classification, Medical Image Analysis, 2024) or the more recent Cellvit++ (Horst et al., Cellvit++: Energy-efficient and adaptive cell segmentation and classification using foundation models, arXiv, 2025) is not a better option here. Authors should better motivate the interest of their method compared with strong segmentation models and provide an experimental validation of their superiority.

- W3: [Major] While the authors claim that standard augmentations from natural-image domain might be harmful in the context of Digital Pathology, to the best of my knowledge, the proposed masking is still combined with standard augmentation from DINOv2. Authors should comment on why standard augmentations are still needed.

- W4: [Major] Related to W3, it is not clear to me why we should expect the “Masked” variant to have such low downstream performance. Could authors elaborate on this? Moreover, the gap between “Standard” and “Mixed” is not very high, in particular at the end of the training. Could this mean that with more training epochs, the gains from the proposed contribution tend to decrease? And also, this seems to invalidate the assumption that standard augmentations are harmful for SSL in Digital Pathology.

- W5: [Minor] I find Figure 5 hard to understand. Could authors provide more details about what conclusions can be drawn from it?

**Questions:**

- Q1: [Related to W1] Can authors perform a quantitative study to better present the absolute performance of their method and also the gain compared with ADIOS?
- Q2: [Related to W2] Can authors motivate the interest of their method compared with strong segmentation models and provide an experimental validation of their superiority (e.g. compared with Cellvit or Cellvit++)?
- Q3: [Related to W3] Can authors comment on why standard augmentations from DINOv2 are still needed in their method?
- Q4: [Related to W4] Could authors elaborate on the low performance of the “Masked” variant which does not seem obvious to me?
- Q5 [Related to W4] Could the small gap between “Standard” and “Mixed” mean that with more training epochs, the gains from the proposed contribution tend to decrease? Does this invalidate the assumption that standard augmentations are harmful for SSL in Digital Pathology?
- Q6: [Related to W5] Could authors provide more details about what conclusions can be drawn from Figure 5?

---

### Official Review · Reviewer_EN5y · 2025-10-31

**Soundness:** 2
**Presentation:** 2
**Contribution:** 2
**Rating:** 4
**Confidence:** 4

**Summary:**

this work deals with self-supervised segmentation (SSS) in histology images.
authors propose to use tissue as a prior for augmentation.
in particular, they used ADIOS model as a mask generator. then, these masks are used as augmentation in DINO model.
results are reported on 6 datasets with different cases.

**Strengths:**

- the writing is good.
- the paper tackles an important topic which is self-supervised segmentation in histology.
- results are provided.

**Weaknesses:**

- limited novelty. the paper covers poorly existing works. it does not state what are the issues/limitations of existing works. authors picked augmentation. but how relevant it is to SSS? what it has to do with SSS? augmentation can be used in any other task. why it is relevant in SSS is not clear. authors simply took existing models and applied them to histology. this limits the contribution and novelty of this work.
in addition, there is not comparison to previous and SOTA works in SSS. it is not clear how this method compares to them.

the writing could be improved. introduction does present a clear problem in SSS. no clear coverage of gaps and how this work can help. proposed method section 3 is a mix between method and implementation details. it is not relevant at that stage to discuss implementation details. it should be left to the experimental section.

**Questions:**

- please improve the writing. clarify the problem for SSS. and position well your method. compare to SOTA in histology.

**Details Of Ethics Concerns:**

none.

---

### Official Review · Reviewer_kWNq · 2025-10-31

**Soundness:** 3
**Presentation:** 3
**Contribution:** 3
**Rating:** 6
**Confidence:** 4

**Summary:**

The author propose a novel data augmentation method tailored to histopathology based on a variant of tissue semantic mask generator (ADIOS). To incorporate this generator into standard data augmentation process in DINO, the authors demonstrate a mixed strategy that achieves the best performance on seven downstream tasks. Further patch token analysis and visualization also show compelling results of the quality of the learn embedding.

**Strengths:**

- Originality: The proposed ADIOS adaption is clever and well-motivated. Instead of random masking,  the authors introduce a reconstruction phase in ADIOS to force the semantically meaningful mask generation process ( cell nuclei, cell borders, vascular
structures)
- Solid experiment results: I appreciate the author's effort for doing sufficient experiments to create 95% confidence interval of model performance along the training iteration for all the tasks. Seeing the model's performance is better across all checkpoints is more convincing than just randomly picking a final one.
- Compelling visualization: The author created many visualization to justify the embedding quality from their approach, which make the approach more convincing
- Detail ablation: the author also attempted to demonstrated their approach work in different architecture (Dino v1, v2)

**Weaknesses:**

- The performance boost seems to be limited: Judging from table 2 and Figure 3, I find the main performance advantage is slide-level task. The results of patch classification look quite competitive to the standard approach
- The importance of base model seems to outweigh the proposed augmentation approach. If we look Figure 6,7,9,10-14. The proposed data augmentation lead to almost no performance gain compared to the standard approach in Dino 1 (Figure 6,7). Similarly, there is no clear separation of the peak in the hue diagram between mixed and standard augmentation. The separation seems to be enhanced from replacing Dino v1 with Dino v2.
- Clarity: I don't find figure 8 (internal representation metrics part) are discussed or referenced in the main paper, which is a bit confusing

**Questions:**

Q1: Do the authors have any comment or thought about why there is more gain in slide-level task than patch level one? If possible, it would be nice to see more evidence of slide-level benchmarks (e.g.,subtyping, survival ..etc)

Q2: What is your thought about the performance gain only become clear when using Dinov2

---

### Note · Authors · 2025-11-17

**Comment:**

We thank the reviewers for time they have put into reviewing our work, we have found several areas of improvement. We believe that this work requires more time before it is acceptable as a publication, and therefore we have decided to withdraw and reformulate our work so that the message is clearer to the reviewers in a future submission. We once again thank the reviewers for their efforts in improving our work.

**Withdrawal Confirmation:**

I have read and agree with the venue's withdrawal policy on behalf of myself and my co-authors.